# Corrupted Image Modeling for Self-Supervised Visual Pre-Training

**Yuxin Fang** [1,2]*  **Li Dong** [2]  **Hangbo Bao** [2]  **Xinggang Wang** [1]†  **Furu Wei** [2]

[1] School of EIC, Huazhong University of Science & Technology  [2] Microsoft Research
{yxf,xgwang}@hust.edu.cn

## Abstract

We introduce **C**orrupted **I**mage **M**odeling (CIM) for self-supervised visual pre-training. CIM uses an auxiliary generator with a small trainable BEiT (Bao et al., 2021) to corrupt the input image instead of using artificial [MASK] tokens, where some patches are randomly selected and replaced with plausible alternatives sampled from the BEiT output distribution. Given this corrupted image, an enhancer network learns to either recover all the original image pixels, or predict whether each visual token is replaced by a generator sample or not. The generator and the enhancer are simultaneously trained and synergistically updated. After pre-training, the enhancer can be used as a high-capacity visual encoder for downstream tasks. CIM is a general and flexible visual pre-training framework that is suitable for various network architectures. For the first time, CIM demonstrates that both ViT and CNN can learn rich visual representations using a unified, non-Siamese framework. Experimental results show that our approach achieves compelling results in vision benchmarks, such as ImageNet classification and ADE20K semantic segmentation.

## 1 Introduction

Vision Transformers (ViTs) (Dosovitskiy et al., 2020) are transferring the landscape of computer vision, not only in terms of the network architecture design, but also the self-supervised pre-training recipe. Masked image modeling (MIM) (Bao et al., 2021), which randomly masks out some input tokens and then recovers the masked content by conditioning on the visible context, is able to learn rich visual representations and shows promising performance on various vision benchmarks (Zhou et al., 2021; He et al., 2021; Xie et al., 2021; Dong et al., 2021; Wei et al., 2021).

Originated in masked language modeling (Devlin et al., 2019), MIM (Figure 1a) is tailor-made for specific architectures (Vaswani et al., 2017), which is generally capable of receiving and processing tokenized inputs such as the artificial [MASK] tokens. Meanwhile, the more common and natural input signal in computer vision is the image in RGB domain with 2D regular grid structures. In order to apply MIM pre-training for images, ViT has to "patchify" the input image into a 1D sequence of non-overlapping patch embeddings, and then use [MASK] tokens to perturb them.

MIM is tightly coupled with the Transformer family, and the usage of [MASK] tokens limits its scope of application to some extent. More importantly, MIM is not directly suitable for convolutional neural networks (CNNs) (LeCun et al., 1989), the dominant architecture for computer vision in the last decade. Introducing [MASK] tokens in any intermediate stage of CNN is infeasible, as convolution's intrinsic dense-sliding-window paradigm causes information leakage between visual features in previous layers and therefore impedes the MIM. Therefore the large CNN family cannot directly benefit from the upsurge of this new pre-training scheme. Moreover, the usage of [MASK] tokens causes a discrepancy between pre-training and fine-tuning (Devlin et al., 2019; Clark et al., 2020), as the artificial [MASK] tokens never appear in the fine-tuning stage.

In this paper, we present a new visual pre-training framework, called **C**orrupted **I**mage **M**odeling (CIM, Figure 1b), which avoids directly manipulating [MASK] tokens on pre-trained models and generalizes quite well to both ViT and CNN architectures. Rather than directly using artificial [MASK] tokens to corrupt a portion of non-overlapping patch embeddings as in MIM, CIM uses

---

*Contribution during internship at Microsoft Research. †Corresponding author.

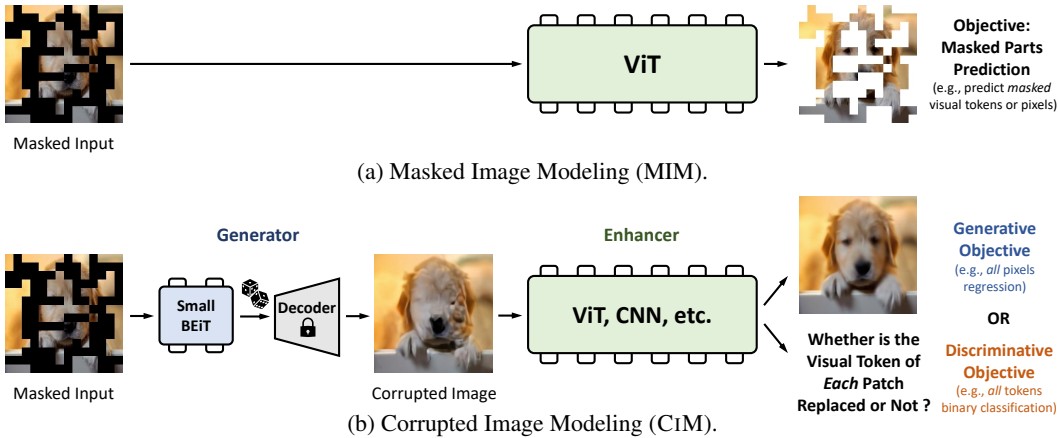

Figure 1: **Overview of our Corrupted Image Modeling (CɪM)** and comparisons with Masked Image Modeling (MIM). MIM (Figure 1a) requires the pre-trained architecture to receive and process the artificial [MASK] tokens, while CɪM (Figure 1b) relaxes these restrictions by using a trainable generator to sample corrupted images serving as the input for the enhancer. Similar to BEiT, the small generator learns to predict the golden visual token produced by the pre-trained frozen image tokenizer encoder (not shown in the figure) based on partial observations of the input. The enhancer can be various architectures including CNN and learns either a generative or a discriminative visual pre-training objective. After pre-training, we throw out the generator and fine-tune the enhancer on downstream tasks. The dice icon in Figure 1b refers to the visual tokens' stochastic sampling process, and the lock icon means the pre-trained image tokenizer decoder is frozen.

a small trainable BEiT (Bao et al., 2021) as an auxiliary generator to corrupt the input image. Specifically, the BEiT generator learns to predict visual tokens at the masked positions, where we utilize the predicted distribution to sample visual tokens' replacements. The replaced visual tokens together with the golden tokens that directly produced by a pre-trained frozen image tokenizer encoder (*e.g.*, the DALL-E (Ramesh et al., 2021) dVAE encoder) given the same input as the small trainable BEiT are then mapped back to the image RGB domain by a pre-trained frozen tokenizer decoder (*e.g.*, the DALL-E dVAE decoder). The resulting corrupted image serves as the input of the enhancer, which is the model to be pre-trained and transferred.

For the enhancer, the choice of pre-training objectives is quite flexible. We study two representatives: a generative objective that regresses *all* the original image pixels given the corrupted image (Dosovitskiy et al., 2020; Chen et al., 2020a), dubbed as **Pix**el **Res**idual learning (RESPIX), and a discriminative objective that predicts whether *each* visual token is replaced by the small generator or not (Clark et al., 2020), dubbed as **Re**placed **V**isual token **Det**ection (REVDET). After pre-training, the enhancer can be used as a strong feature extractor for visual downstream tasks.

Overall, CɪM is a general and flexible pre-training framework suited for different kinds of visual encoders. For the first time, we demonstrate that both ViT and CNN can learn rich visual representations using a unified *non-Siamese* structure. Experimental results show that our approach achieves compelling results in vision benchmarks, such as ImageNet classification and ADE20K semantic segmentation. We hope CɪM can serve as a promising starting point for exploring flexible & unified visual representation learning of various architectures.

## 2 CORRUPTED IMAGE MODELING (CɪM)

Figure 1b shows the overview of CɪM. Our approach simultaneously learns two neural networks: an auxiliary *generator* and an *enhancer*. The generator is used to corrupt the input image, while the enhancer receives the corrupted image (Figure 2) and learns either a generative or a discriminative visual pretext task. After pre-training, we throw out the generator and fine-tune the enhancer on downstream tasks.

### 2.1 GENERATOR

Rather than using artificial [MASK] tokens to corrupt the input image, we learn a trainable auxiliary **generator** to relax the architectural constraints of MIM. Moreover, the generator enriches the diversity

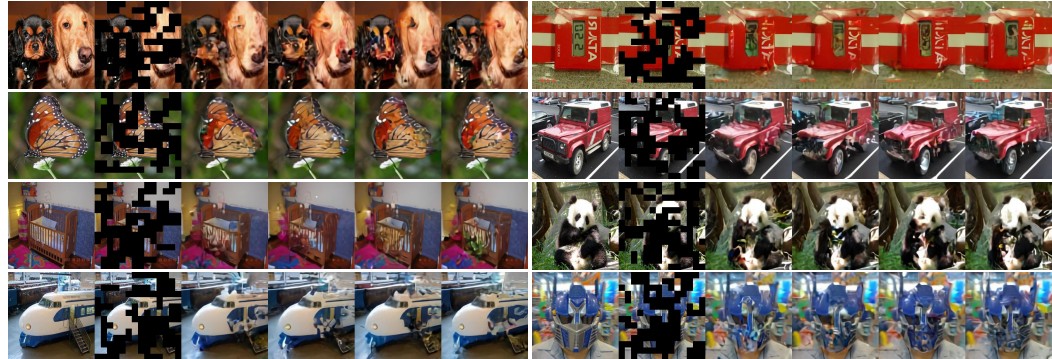

(a) Corrupted image samples from *ImageNet-1K training set*. Although the model is trained using the same dataset, the corrupted image samples still vary to a certain extent. Therefore during pre-training, the generator is able to continuously provide abundant and diverse corrupted samples for the enhancer.

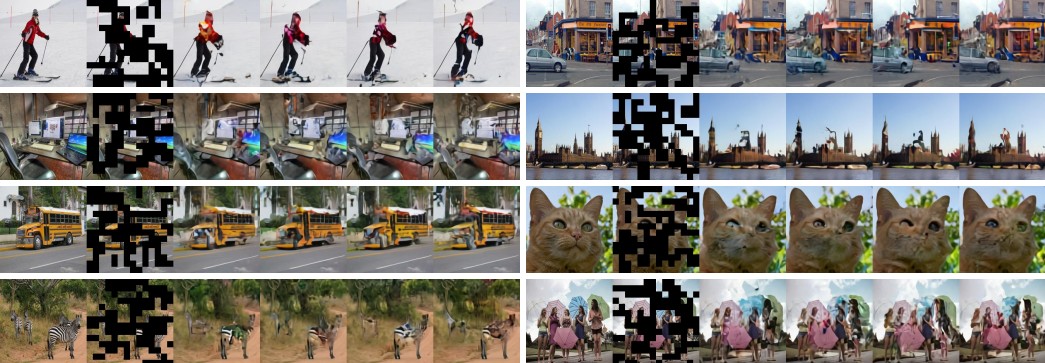

(b) Corrupted image samples from *COCO val split* (Lin et al., 2014) using ImageNet-1K pre-trained model.

Figure 2: **Visualizations of some corrupted image samples.** For each image set, we show (from left to right) the original image, the masked image, and four different corrupted images sampled from the generator output distribution with the *same* masked input. Simple stochastic sampling can greatly enrich the corrupted image distribution in terms of both low-level features and high-level semantics, which feeds the enhancer better.

of corrupted images via stochastic sampling, which helps the enhancer generalize. The generator consists of a pre-trained frozen image tokenizer, and a small trainable BEiT (Bao et al., 2021).

**The frozen image tokenizer** in CIM is a pre-trained discrete variational autoencoder (dVAE) (Rolfe, 2016; Van Den Oord et al., 2017), consisting of a paired encoder and decoder. The tokenizer encoder maps the input image into a sequence of discrete visual tokens with a fixed vocabulary size. The tokenizer decoder can recover semantically plausible images given a permutation of appropriate and meaningful visual tokens. We directly use the DALL-E (Ramesh et al., 2021) tokenizer, following BEiT.

**The small BEiT** consists of several Transformer encoder layers and is trained to perform MIM, which uses two views for each input image, *i.e.*, a sequence of non-overlapping patch embeddings, and their corresponding discrete visual tokens. Patch embeddings are linearly embedded from non-overlapping input image patches. Discrete visual tokens are from the DALL-E tokenizer encoder, serving as the prediction target for BEiT.

Given a sequence of patch embeddings, the small BEiT randomly masks out a set of positions. The patch embeddings at the masked positions are replaced with special mask embeddings. The small BEiT takes this corrupted sequence of patch embeddings as the input, and learns to predict the corresponding discrete visual tokens at all masked positions given the visible context only. In CIM pre-training, the size of the small BEiT we use is typically a quarter or a half of the enhancer.

Using discrete visual tokens to represent images enables CIM to perform *stochastic sampling* during the corrupted image's generation process, which greatly enriches the output set of the generator. In this paper, we directly sample from softmax with a temperature of 1 at all the masked positions according to the small BEiT output distribution. All the masked tokens are replaced by the sampled visual tokens. The sampled tokens together with the golden tokens that are directly produced by the image tokenizer encoder at all the non-masked positions constitute the input for the image tokenizer decoder. Then the decoder maps those plausible visual tokens to a *corrupted image* (refer to examples in Figure 2), which serves as the input for the enhancer.

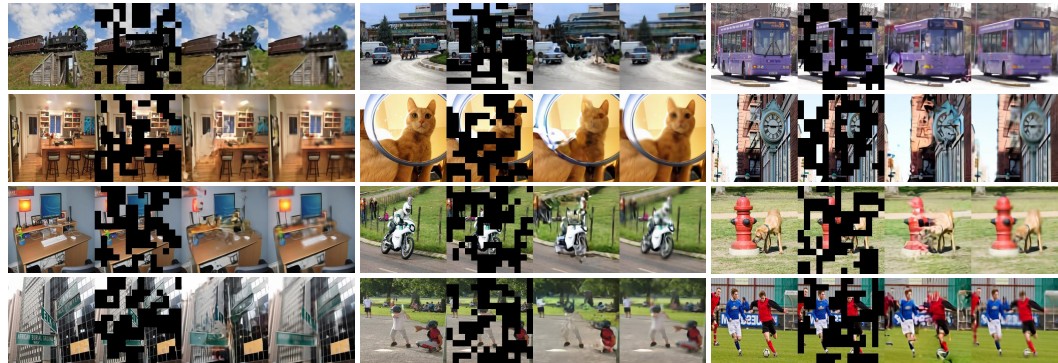

(a) CIM-RESPIX pre-training objective with *sliding window normalized* pixels as the enhancer prediction target.

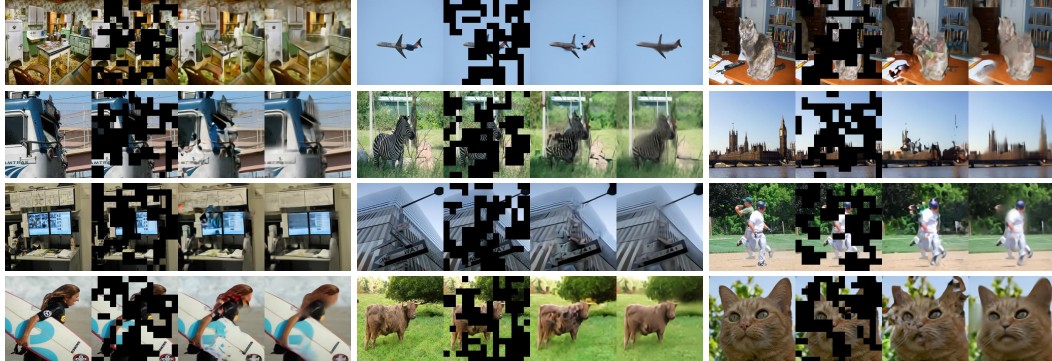

(b) CIM-RESPIX pre-training objective with *unnormalized* pixels as the enhancer prediction target.

Figure 3: **Example visualization results on COCO val split images** from vanilla ViT-Base/16 model pre-trained with the **RESPIX** objective using ImageNet-1K training data. For each image quadruplet, we show the original input image (1st column), the masked input image for the generator (2nd column), the corrupted image sampled from the generator output (3rd column), and the enhancer output (4th column). Given the corrupted image, the enhancer is able to perform image denoising, deblurring and completion, *etc.*, and learns to predict plausible output in terms of both low-level features as well as high-level semantics.

## 2.2 ENHANCER

Given the corrupted image sampled from the auxiliary generator, the enhancer learns either a generative or a discriminative visual pretext task. The prediction head is a simple linear layer, and the choice of pre-training objectives is quite flexible. In this paper, we study two representative objectives, coined as **Pix**el **Res**idual learning (RESPIX) and **Re**placed **V**isual token **Det**ection (REVDET).

**RESPIX** (Figure 3) is a generative pretext task that requires the enhancer to predict the uncorrupted pixel value for *all* positions given the corrupted input. Instead of directly regressing the original pixel, MAE (He et al., 2021) suggests learning the normalized counterpart. Specifically, the image is partitioned into a set of non-overlapping patches, and each pixel is normalized by the mean and standard deviation of all pixels in the patch it lives in, *i.e.*, patches with layer normalization (Ba et al., 2016) are the reconstruction target.

In CIM, we further propose to normalize the prediction target inside a *sliding* window, *i.e.*, each pixel is normalized by all pixels in a local $8 \times 8$ sized window centered at where the target pixel lives in. We observe improved representation quality using the sliding window normalization paradigm.

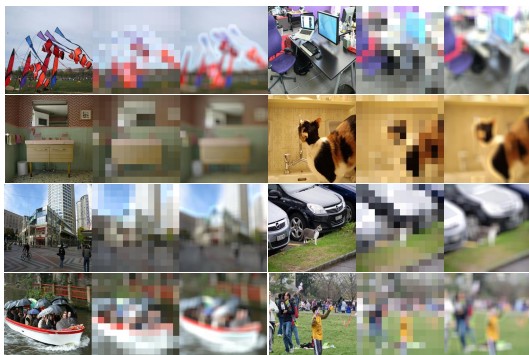

Figure 4: **Normalizations as learning templates for RESPIX**. For each image triplet, we visualize the original image (left), the template of using non-overlapping window normalization (He et al., 2021), and the template of the proposed sliding window normalization paradigm. Our approach can provide more accurate and *moderate* hints that can boost the enhancer's pre-training as well as improve its representation quantity.

Naive pixel recovery without normalization tends to waste modeling capability on learning short-range dependencies and high-frequency details (Ramesh et al., 2021; Bao et al., 2021), while the normalized target can mitigate irrelevant information fittings. From another perspective, normalizations are equal to providing learning templates, as shown in Figure 4. With the normalized prediction target, the enhancer only needs to learn the *residual* pixel value at each position given the normalized pixel value, while the unnormalized target provides no hint therefore the enhancer has to "learn to see in the dark" (*i.e.*, regress from RGB: 0, 0, 0). It is also hard for the enhancer to learn without a template since the corrupted image usually provides bad priors (refer to the corrupted image samples in Figure 2 and Figure 3). Therefore, we believe appropriate and moderate hints will help the enhancer see better.

**REVDET** is a discriminative visual pretext task that requires the enhancer to determine whether *each* visual token is replaced by a generator sample or not. To be specific, the visual tokens produced by the pre-trained frozen image tokenizer encoder are considered as golden tokens. If a generated visual token is different from the golden token at the same position, that generated token is considered "replaced", and vice versa.

REVDET is inspired by ELECTRA (Clark et al., 2020) in language modeling. The main difference is, in the proposed CIM, the determining criterion of replacement is hidden in the corrupted image. Token replacement is a kind of local, high-frequency operation by nature. However, the visual token set after sampling and replacement is further smoothed and processed by the image tokenizer decoder. Therefore the token sampling and replacement operations are finally embodied as non-local, high-level semantics changes in the corrupted image. The enhancer is required to "decrypt" it and identify all the replaced tokens given the corrupted input, which yields a nontrivial and meaningful visual pretext task[1]. To some extent, REVDET also learns the DALL-E dVAE's visual codebook similar to BEiT, but in a discriminative manner.

**The enhancer** is regarded as the visual encoder after pre-training. Moreover, unlike masked image modeling, CIM does not assume too many *architectural priors* for the pre-trained network. We successfully pre-train a high-capacity vanilla ResNet-50 (He et al., 2016), ResNet-50x2 and ResNet-50x4 enhancers that achieve compelling transfer learning performance using a similar configuration as pre-training a ViT enhancer. For the first time, we demonstrate that both ViT and CNN can learn strong visual representations using a unified *non-Siamese* framework.

## 2.3 Training and Optimization

The auxiliary generator and the enhancer are simultaneously trained and *synergistically* (rather than *adversarially* as GAN (Goodfellow et al., 2014)) updated. The trainable part of the generator, *i.e.*, the small BEiT, learns a MIM objective in the same vein as in (Bao et al., 2021). The whole pre-trained image tokenizer is frozen.

For the RESPIX visual pretext task, the enhancer is optimized by a combination of $l_1$ and $l_2$ loss. For the REVDET visual pretext task, the enhancer is learned by binary cross-entropy loss. Notice that the gradients of the enhancer are not back-propagated through the generator. A detailed formulation is presented in Appendix A.3.

## 3 Experiments

We study CIM self-supervised pre-trained vanilla ViT-Small/16 (Touvron et al., 2021a), vanilla ViT-Base/16 (Dosovitskiy et al., 2020) and vanilla ResNet-50 (He et al., 2016) models. We use the actual processed images / views to measure the pre-training epochs (PT epochs). ImageNet-1K (Deng et al., 2009) training data is used to pre-train the small BEiT and the enhancer. Our pre-training setting generally follows BEiT (Bao et al., 2021). Unlike BEiT, CIM only uses cropping and flipping for data argumentation, while dropout (Srivastava et al., 2014) and stochastic depth (Huang et al., 2016) are not applied. The detailed pre-training settings are summarized in the Appendix A.4. Notably, the pre-training configurations are *almost the same* for both ViT and CNN architectures.

In order to evaluate the pre-trained representations from CIM, for both ViT and CNN architectures, we conduct supervised end-to-end fine-tuning (FT) experiments on ImageNet-1K (Deng et al., 2009) image classification in §3.1, and ADE20K (Zhou et al., 2019) semantic segmentation in §3.2.

---

[1]Therefore, REVDET can be also interpreted as "**Rev**erse token **Det**ection from corrupted image".

Table 1: ImageNet-1K end-to-end fine-tuning top-1 accuracy of vanilla ViT-Small/16 and ViT-Base/16 models.
†Doubled attention heads.     ‡Our reproduction.

| Models | PT Epochs | Top-1 |
|---|---|---|
| *ViT-Small/16 model results* | | |
| Scratch (Touvron et al., 2021a) | | 79.9 |
| MoCo-v3† (Chen et al., 2021) | 600 | 81.4 |
| DINO (Caron et al., 2021) | 1600 | 81.5 |
| BEiT (Bao et al., 2021) | 300 | 81.3 |
| **CIM-RESPIX (Ours)** | 300 | 81.5 |
| **CIM-REVDET (Ours)** | 300 | **81.6** |
| *ViT-Base/16 model results* | | |
| Scratch (Touvron et al., 2021a) | | 81.8 |
| Scratch (He et al., 2021) | | 82.3 |
| DINO (Caron et al., 2021) | 1600 | 82.8 |
| MoCo-v3 (Chen et al., 2021) | 600 | 83.2 |
| BEiT (Bao et al., 2021) | 300 | 82.9 |
| BEiT (Bao et al., 2021) | 800 | 83.2 |
| MAE‡ (He et al., 2021) | 800 | 83.1 |
| **CIM-REVDET (Ours)** | 300 | **83.3** |
| **CIM-RESPIX (Ours)** | 300 | **83.3** |

Table 2: ImageNet-1K end-to-end fine-tuning top-1 accuracy of vanilla ResNet-50 model. RSB (Wightman et al., 2021) is the current vanilla ResNet state-of-the-art training procedure.

| Models | PT Epochs | Top-1 |
|---|---|---|
| *Fine-tuning for 100 epochs* | | |
| RSB A3 (Wightman et al., 2021) | | 78.1 |
| **CIM-REVDET (Ours)** | 300 | **78.8** |
| *Fine-tuning for 300 epochs* | | |
| RSB A2 (Wightman et al., 2021) | | 79.8 |
| SimSiam (Chen & He, 2021) | 400 | 79.1 |
| MoCo-v2 (Chen et al., 2020c) | 400 | 79.6 |
| SimCLR (Chen et al., 2020b) | 800 | 79.9 |
| SimCLR (Chen et al., 2020b) | 2000 | 80.0 |
| BYOL (Grill et al., 2020) | 400 | 80.0 |
| SwAV (Caron et al., 2020) | 600 | 80.1 |
| **CIM-RESPIX (Ours)** | 300 | 79.9 |
| **CIM-REVDET (Ours)** | 300 | **80.5** |
| *Fine-tuning for 600 epochs* | | |
| RSB A1 (Wightman et al., 2021) | | 80.4 |
| **CIM-REVDET (Ours)** | 300 | **80.7** |

Ablation study on ImageNet-1K is presented in §3.3. For ImageNet-1K, we observe ∼0.2 Top-1 acc. fluctuations. For ADE20K, we observe ∼0.5 mIoU fluctuations. We report key results using the median of 3 independent runs.

## 3.1  IMAGE CLASSIFICATION

**ViT.**   The ImageNet-1K end-to-end fine-tuning top-1 accuracy of vanilla ViT-Small/16 and ViT-Base/16 models are presented in Table 1. We fine-tune the small-sized model for 200 epochs, and the base-sized model for 100 epochs. Other self-supervised methods in Table 1 use the same or longer fine-tuning schedule. The fine-tuning hyperparameters mostly follow BEiT, while our layer-wise lr decay rate is set to 0.8 as suggested by Clark et al. (2020). See Appendix A.4 for detailed configurations.

As shown in Table 1, CIM is able to achieve better accuracy with fewer pre-training epochs compared with other representative self-supervised vanilla ViT models. Moreover, we find both REVDET and RESPIX visual pretext task can help the ViT enhancer learn useful representations.

**ResNet-50.**   We demonstrate that CIM can also pre-train a high-capacity ResNet-50 model with the fewest possible modifications from the ViT pre-training settings that can achieve compelling fine-tuning performances on ImageNet-1K. We use the AdamW optimizer (Loshchilov & Hutter, 2017) for fine-tuning, and other configurations basically follow the advanced training recipe of RSB (Wightman et al., 2021). For other self-supervised baselines, we select the best lr out of {5e-3, 8e-3, 12e-3} and keep other settings unchanged to ensure a fair and challenging competition. The detailed configurations are given in Appendix A.4.

As shown in Table 2, under such a demanding training procedure, CIM pre-trained ResNet-50 model can still outperform several representative self-supervised methods based on the Siamese framework as well as the modernized state-of-the-art ResNet-50 results. Using the improved fine-tuning recipe, we also observe performance degeneration for some self-supervised baselines compared with the RSB from scratch results. Notably, even with the extreme 600-epoch training schedule, the CIM representation can still improve the state-of-the-art RSB A1 by 0.3%.

## 3.2  SEMANTIC SEGMENTATION

We study the transfer learning performance of CIM pre-trained vanilla ViT-Base/16 and ResNet-50 models on the ADE20K semantic segmentation benchmark. The pre-trained models are used as an encoder, and we purposefully choose *simple decoders* to better reveal the pre-trained representations. Experiments are based on the code of Bao et al. (2021); MMSegmentation (2020).

Specifically, for ViT-Base/16 we use a simple linear layer as the decoder, and for ResNet-50 we choose the ubiquitous FCN (Long et al., 2015) as the decoder. For ViT, the baseline settings as well as the fine-tuning recipes are from (Bao et al., 2021). We select the best lr out of {1e-4, 3e-4, 5e-4, 7e-4} for DINO. For BEiT we use the default setting (lr 7e-4 with a decay rate of 0.65). For CIM pre-trained ViT, we set the fine-tuning lr equal to 3e-4 with a decay rate of 0.8 as suggested by Clark et al. (2020). For ResNet-50, we use the canonical configuration for all methods, *i.e.*, the optimizer is SGD with a momentum of 0.9, lr follows a poly decay schedule, and the batch size is 16. The training crop size is set to 512 for all models, and we use single-scale inference.

As summarized in Table 3, when transferred to semantic segmentation task, CIM pre-trained models can still achieve competitive performances compared with other approaches. Notably, for ResNet-50, as the fine-tuning schedule becomes longer (*i.e.*, 80k iterations → 160k iterations), the performance gain from the ImageNet-1K supervised pre-trained representation is small. Moreover, the performance is even worse than training from scratch. Meanwhile, the CIM pre-trained ResNet-50 representation can provide sustaining performance gain for a longer fine-tuning schedule.

Table 3: ADE20K semantic segmentation performances (mIoU) of ViT and ResNet-50 models.

| Models | PT Epochs | Top-1 |
|---|---|---|
| *Fine-tuning for 160k iterations* | | |
| DINO (Caron et al., 2021) | 1600 | 43.0 |
| BEiT (Bao et al., 2021) | 300 | 43.2 |
| **CIM-RESPIX (Ours)** | 300 | **43.5** |
| **CIM-REVDET (Ours)** | 300 | **43.6** |

(a) Vanilla ViT-Base/16 as encoder with one *linear layer* as decoder.

| Models | PT Epochs | mIoU |
|---|---|---|
| *Fine-tuning for 80k iterations* | | |
| Training from Scratch | | 29.9 |
| IN1K Supervised[†] (He et al., 2019) | 120 | 35.9 |
| **CIM-REVDET (Ours)** | 300 | **36.2** |
| *Fine-tuning for 160k iterations* | | |
| Training from Scratch | | 36.7 |
| IN1K Supervised (He et al., 2019) | 120 | 36.1 |
| BYOL (Grill et al., 2020) | 400 | 37.1 |
| SimSiam (Chen & He, 2021) | 400 | 37.1 |
| SwAV (Caron et al., 2020) | 600 | 37.2 |
| MoCo-v2 (Chen et al., 2020c) | 400 | 37.5 |
| SimCLR (Chen et al., 2020b) | 800 | 37.6 |
| SimCLR (Chen et al., 2020b) | 2000 | 37.7 |
| **CIM-RESPIX (Ours)** | 300 | **38.7** |
| **CIM-REVDET (Ours)** | 300 | **39.0** |

(b) Vanilla ResNet-50 as encoder with a classic FCN as decoder.

Together with the observation from §3.1, we demonstrate CIM is a general, non-Siamese framework that is capable of pre-training both strong ViT and CNN visual encoders.

## 3.3 ABLATION STUDIES

Ablation studies are conducted using 300-epoch CIM-RESPIX pre-trained ViT-Base model with 100 epochs fine-tuning on ImageNet-1K unless specified. Some additional analysis is available in Appendix A.1.

**Masking Strategy and Masking Ratio.** As shown in Table 4, we observe CIM works better with simple random masking (He et al., 2021; Xie et al., 2021) compared with the blockwise masking strategy (Bao et al., 2021).

The optimal random masking ratio is around 50%, which we find also holds for the REVDET pretext task, in part because it provides almost equal amounts of positive and negative training samples.

**The Small BEiT Depth and Weight Sharing.** Following Meng et al. (2021); Chi et al. (2021), we adjust the size of the small trainable BEiT by varying its depth (*i.e.*, the number of Transformer encoder layers) instead of its width (*i.e.*, the feature dimension). As summarized in Table 5, the small BEiT with 4 to 6 layers is generally fine.

It is also beneficial to share the patch embedding layer as well as the first two Transformer encoder layers between the small BEiT and enhancer as long as the enhancer is also ViT. We hypothesize that sharing the earlier layers can help calibrate the enhancer since the small BEiT receives the real inputs while the enhancer sees the same sources but with corrupted views.

Table 4: Ablation study: masking strategy and masking ratio.

| Masking Strategy | Masking Ratio | Top-1 Acc. |
|---|---|---|
| Blockwise | 40% | 82.8 |
| Blockwise | 50% | 82.9 |
| Blockwise | 60% | 82.8 |
| Random | 40% | 83.0 |
| Random | 50% | **83.3** |
| Random | 60% | 83.1 |

Table 5: Ablation study: depth of the small BEiT in the generator and weight sharing.

| # Enc. Layers | Weight Sharing | Top-1 Acc. |
|---|---|---|
| 4 | ✗ | 83.1 |
| 4 | ✓ | **83.3** |
| 5 | ✓ | 83.2 |
| 6 | ✓ | 83.2 |
| 7 | ✓ | 83.1 |
| 8 | ✓ | 82.9 |

Table 6: Ablation study: pixel reconstruction target for RESPIX pre-training objective.

| RESPIX Recon. Target | Top-1 Acc. |
|---|---|
| w/o norm. | 82.8 |
| norm. w/ non-overlap win. | 83.0 |
| norm. w/ sliding win. | **83.3** |

Table 7: Ablation study: sampling strategy for visual tokens.

| Sampling Strategy | Top-1 Acc. |
|---|---|
| Uniform sampling | 77.2 |
| `argmax` sampling | 78.5 |
| `softmax` sampling | **83.3** |

**Target for RESPIX.**    We believe an appropriate normalization technique can provide moderate hints that can help improve the enhancer's representation quality with the RESPIX visual pretext task (see our discussion of Figure 4). As shown in Table 6, the proposed sliding window normalization improves the fine-tuning accuracy by 0.5% *vs.* the reconstruction target without normalization, and is also 0.3% better than the normalization method proposed in He et al. (2021).

**Sampling Strategy for Visual Tokens.**    Using discrete visual tokens to represent images enables CIM to use stochastic sampling techniques during the corrupted image's generation process, which can greatly enrich the output set of the generator and help the enhancer generalize well. For masked image modeling, randomly masking out a portion of patch embeddings can help regularize the pre-training, while for our approach, regularization for the enhancer mainly comes from the diversity of the corrupted images, therefore regularizations such as dropout & droppath are not used in CIM.

As presented in Table 7, the visual token representation with simple stochastic sampling from the generator output distribution is crucial for CIM. In contrast, we find that uniform sampling from the codebook of the image tokenizer regardless of the generator distribution or `argmax` sampling from the distribution cannot provide meaningful or diverse samples and therefore fails to pre-train the enhancer as expected.

**Image Corrupting Strategy.**    We find that it is crucial to use a generator with a small trainable BEiT to corrupt images in order to successfully pre-train CNN with the proposed CIM. We experiment with another generative visual pretext task for ResNet-50 pre-training, *i.e.*, using 50% random erasing (Zhong et al., 2020) to corrupt the input image, and the model is required to recover the erased pixels based on the visible context. We find this pretext task fails to transfer well. A parallel work Tian et al. (2022) also finds that only using hand-crafted transformations to corrupt images is not quite satisfactory in generative visual pre-training of ViT.

**Scaling CIM to Larger CNNs.**    We study the scaling behavior of our CIM to larger CNNs. We choose two popular architectures in self-supervised learning literature: ResNet-50x2 and ResNet-50x4 (with width multipliers of 2x and 4x of vanilla ResNet-50, respectively), and study the end-to-end fine-tuning performance on ImageNet-1K in Table 8. We use an improved training recipe following Touvron et al. (2021a); Wightman et al. (2021), therefore our from

Table 8: Scaling CIM pre-training to larger ResNet.

| Methods | PT Epochs | FT Epochs | Top-1 Acc. |
|---|---|---|---|
| *ResNet-50x2 (#params: 94M)* | | | |
| From Scratch | - | 400 | 81.1 |
| SimCLR (Chen et al., 2020b) | 1000 | 100 / 200 | 81.6 / 82.1 |
| **CIM-REVDET (Ours)** | 300 | 100 / 200 | 81.7 / **82.2** |
| *ResNet-50x4 (#params: 375M)* | | | |
| From Scratch | - | 400 | 80.9 |
| SimCLR (Chen et al., 2020b) | 1000 | 100 | **82.6** |
| SimMIM (Xie et al., 2021) | 300 | 100 | 81.6 |
| **CIM-REVDET (Ours)** | 300 | 100 | **82.6** |

scratch and SimCLR baselines are much higher (∼2 points higher) than the original results in Chen et al. (2020b). Notice that it is non-trivial to pre-train those large CNNs (*e.g.*, ResNet-50x4 is 14 times bigger than ResNet-50 in #params). Under the end-to-end fine-tuning protocol, CIM is better than

the recent MIM-based approach SimMIM and competitive with the representative Siamese model SimCLR.

**Scaling CIM to Larger ViT.** We study the scaling behavior of our CIM to ViT-Large in Table 9. Indeed, our approach can give ViT-Large a better initialization compared with the random initialization, and can also achieve better performance than MoCo-v3 that based on the canonical Siamese framework. Meanwhile, CIM still lags behind the MIM-based BEiT. Nevertheless, we believe CIM can serve as a promising starting point for exploring unified visual pre-training of various architectures.

Table 9: Scaling CIM pre-training for ViT-Large.

| Methods | Top-1 Acc. |
|---|---|
| *ViT-Large (#params: 304M)* | |
| From Scratch (He et al., 2021) | 82.6 |
| MoCo-v3 (Chen et al., 2021) | 84.1 |
| BEiT (Bao et al., 2021) | **85.2** |
| **CIM-RESPIX (Ours)** | 84.3 |

**Limitation and Discussion.** The image corrupting process of CIM still has a large room for improvement, which determines the characteristics and styles of the corrupted image distribution. The tokenizer we currently use is essentially a large CNN and adds nontrivial overhead during pre-training, *i.e.*, the wall-clock time of 1-epoch training is about $2\times$ of BEiT. Other image tokenizers, such as ViT-VQGAN (Yu et al., 2021), which report much higher throughput and better generation quality, deserve an in-depth study for CIM pre-training in the future.

## 4 RELATED WORK

**Siamese Framework** is the dominating self-supervised visual pre-training approach over the past few years, which typically relies on strong hand-crafted data augmentations to generate different views of the same image and learns in a contrastive manner. To maintain a large and informative negative sample set, memory banks (He et al., 2020) or large batch size (Chen et al., 2020b) is used. Follow-ups (Grill et al., 2020; Chen & He, 2021) further eliminate the requirement of using negative samples. Recent works (Caron et al., 2021; Chen et al., 2021) study self-supervised visual pre-training of ViT within Siamese frameworks.

**Masked Image Modeling (MIM)** learns rich visual representations via masked parts prediction by conditioning on visible context only. ViT (Dosovitskiy et al., 2020) and iGPT (Chen et al., 2020a) report the first meaningful MIM visual pre-training results. BEiT (Bao et al., 2021) greatly improves MIM's performance via masked visual token prediction, and PeCo (Dong et al., 2021) finds injecting perceptual similarity during visual codebook learning benefits MIM pre-trained representation. Recent work (He et al., 2021; Xie et al., 2021; Wei et al., 2021) re-explore pixel / feature regression in MIM, while Li et al. (2021); Zhou et al. (2021); El-Nouby et al. (2021) incorporate MIM within Siamese frameworks. As MIM is originated in masked language modeling (Devlin et al., 2019), CIM is inspired by Clark et al. (2020). In CIM, visual-token-based MIM plays an important role during the corrupted image generation process, as the stochastic sampling ability greatly enriches the corrupted image set.

## 5 CONCLUSION

We introduce a general self-supervised visual pre-training framework with few architectural constraints for the model to be pre-trained and transferred. Unlike the mainstream Siamese pre-training methods based on strong artificial data augmentations as well as MIM pre-training relying on randomly inserting artificial [MASK] tokens to input embeddings, CIM pre-trained encoder learns from the corrupted view generated from a trainable neural network's output distribution. Given the stochastic sampling ability, CIM defends using discrete visual token representations during pre-training to some extent. Experimental results show that our approach achieves competitive performance on canonical ViT and CNN models. We hope CIM can serve as a promising starting point for exploring flexible & unified visual representation learning of various architectures.

## ACKNOWLEDGMENT

This work is in part supported by the National Key Research and Development Program of China under Grant 2022YFB4500602. We would like to acknowledge Yaru Hao for the helpful discussions.

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

# A APPENDIX

## A.1 ADDITIONAL ANALYSIS

**Relationship between the type of the generator and the performance of the enhancer.** What makes a "good" generator for the enhancer? We believe there are three main factors that affect the output quality of the generator: (1) The masking strategy and masking ratio of the generator's inputs. (2) The size / capacity of the small trainable BEiT. (3) The type of image tokenizers.

While there are many perspectives / ways to evaluate a generator, this study focuses on visual pre-training of the enhancer, so we are particularly interested in how these factors affect the enhancer's fine-tuning performance on downstream visual recognition tasks.

Factor 1 & 2 has already been well studied in Table 4 & Table 5 respectively: either a too "weak" generator (*e.g.*, too much masking or the size of the trainable BEiT is too small) or a too "strong" generator (*e.g.*, too less masking or the trainable BEiT is too large) is harmful to the fine-tuning performance of the enhancer.

As for Factor 3, the image tokenizer represents a given image in the RGB domain as a permutation of discrete tokens with a fixed vocabulary size. This compact representation along with the stochastic sampling process can generate an abundant & diverse input set to feed the enhancer better. However, if the generator is too strong & robust that can always generate near ground truth output regardless of the stochastic sampling, the enhancer can hardly learn useful representations or even be wrongly penalized.

To show that, in Table 10 we study another well-established and open-sourced image tokenizer, VQGAN (Esser et al., 2021), on the ViT-B enhancer with 300 epochs pre-training & 100 epochs fine-tuning on ImageNet-1k. We also study the effects of directly using a MAE-Base model as the generator.

Table 10: Study of different generator tpye of CIM pre-training for ViT-Base.

| Generator Type of CIM | Top-1 Acc. |
|---|---|
| MAE-style generator w/ 50% masking ratio | 82.6 (-0.7) |
| BEiT-Style generator w/ VQGAN tokenizer | 82.9 (-0.4) |
| BEiT-Style generator w/ DALL-E tokenizer (our default setting) | **83.3** |

For the MAE-style generator, we sample RGB color values at all masked positions of the MAE decoder outputs. Since the stochastic sampling is performed on the RGB domain, only some low-level features (mainly color) can be changed and corrupted. Therefore the enhancer only learns to correct low-level attributes.

For the BEiT-style generator w/ VQGAN tokenizer, compared with the DALL-E tokenizer used as default, the VQGAN tokenizer is trained with two additional losses, *i.e.*, the perceptual loss (Zhang et al., 2018) as well as the GAN loss (Isola et al., 2017). These two additional losses are originally intended for high-quality image synthesis, but could make the tokenizer become too strong & robust to generate appropriate corrupted samples for the enhancer. We visualize the corrupted samples from the VQGAN tokenizer, and we find it nearly reconstructs the original input even with stochastic token sampling. Therefore the samples from the VQGAN tokenizer are not diverse enough and cannot provide rich supervision for the enhancer to learn transferable representations.

Overall, it is hard to find a good indicator from the generator that can directly reflect and measure the representation quality of the enhancer. By now, the best way is to honestly fine-tune the pre-trained enhancer on downstream tasks.

**Study of training the generator first and keeping it fixed for the enhancer's pre-training.** We tried first train the generator separately for 300 epochs and then pre-train the enhancer for another 300 epochs while keeping the generator's weights fixed. The performance suffers from a 0.4% degeneration. We hypothesize synergetic & simultaneous training provides a curriculum-like pre-training strategy for the enhancer where the generator starts off weak but gets better throughout training.

**Additional training cost of CIM compared to simple mask prediction with the same mask ratio.**
We study the relationship between the pre-training time and downstream performances of different approaches for both ViTs and ConvNets in Table 11 and Table 12 respectively.

Since CIM is built upon BEiT, we choose BEiT as the masked image modeling baseline approach of ViTs. Here, we first study the ViT-B model's 100-epoch fine-tuning performance on ImageNet-1k val set with different pre-training schedules in Table 11. The wall-clock time of 1-epoch pre-training of CIM is about 1.8x of BEiT (CIM has an additional tokenizer decoder compared with BEiT) on the same machine.

Table 11: Study of the training cost for ViT-Base pre-training.

| Methods | PT Epochs | Relative PT Time | Top-1 Acc. |
|---------|-----------|------------------|------------|
| BEiT    | 300       | 1.0x             | 82.9       |
| BEiT    | 800       | 2.7x             | 83.2       |
| BEiT    | 1600      | 5.3x             | 83.3       |
| CIM     | 800       | 1.8x             | 83.3       |
| CIM     | 800       | 4.8x             | 83.4       |

In Table 12, we also study the ResNet-50x4 model's 100-epoch fine-tuning performance on ImageNet-1k val set with different pre-training schedules. We choose SimMIM as the masked image modeling baseline approach of ConvNets, for it reports the ResNet-50x4 model's result in Appendix E of its paper. The wall-clock time of 1-epoch pre-training of CIM is about 2.6x of SimMIM (CIM has an additional generator, including a small BEiT and a tokenizer encoder & decoder compared with SimMIM) on the same machine.

Table 12: Study of the training cost for ResNet-50x4 pre-training.

| Methods | PT Epochs | Relative PT Time | Top-1 Acc. |
|---------|-----------|------------------|------------|
| SimMIM  | 300       | 1.0x             | 81.6       |
| CIM     | 100       | 0.9x             | 82.2       |

These results imply that CIM can obtain better fine-tuning performance with less pre-training time compared with baseline approaches for both ViTs and ConvNets.

## A.2 A NOTE ON VISUALIZATIONS IN §2.2 AND FIGURE 3

Since there exists information loss in any form of normalization, we have to inject the original image's information in order to visualize the enhancer output (4th column in Figure 3a). In order to comprehensively demonstrate our method's behavior, we also include the unnormalized counterpart in Figure 3b for reference, where there is no additional information injection during visualization.

## A.3 TRAINING AND OPTIMIZATION DETAILS

The auxiliary generator and the enhancer are simultaneously trained and synergistically (rather than adversarially as GAN (Goodfellow et al., 2014)) updated. The trainable part of the generator, *i.e.*, the small BEiT, learns a MIM objective in the same vein as in BEiT (Bao et al., 2021). Formally, given an input image's patch embedding sequence $\boldsymbol{x} = (x_1, ..., x_n)$, we randomly mask $k$ embeddings at positions $\boldsymbol{m} = (m_1, ..., m_k)$ using [MASK] token[2]. The resulting masked input sequence $\boldsymbol{x}^{\text{masked}}$ for BEiT is:

$$m_i \sim \text{uniform}\{1, n\}, \ \text{ for } i = 1, ..., k,$$
$$\boldsymbol{x}^{\text{masked}} = \text{replace}(\boldsymbol{x}, \boldsymbol{m}, \texttt{[MASK]}), \tag{1}$$

where the $\text{replace}(\boldsymbol{x}, \boldsymbol{m}, \texttt{[MASK]})$ operation denotes using the special [MASK] token to replace patch embeddings of $\boldsymbol{x}$ at positions $\boldsymbol{m}$. The small BEiT then encodes $\boldsymbol{x}^{\text{masked}}$ and learns to maximize

---

[2]Typically, we set $k$ equal to $100 \sim 120$ given the input sequence length $n$ of 196, *i.e.*, about $50\% \sim 60\%$ of the total input patch embeddings are masked out.

$\log p_{\text{BEiT}}(\boldsymbol{g} \mid \boldsymbol{x}^{\text{masked}})$, *i.e.*, the log-likelihood of the golden visual tokens $\boldsymbol{g} = (g_1, ..., g_k)$ at the masked positions $\boldsymbol{m}$ conditioned on $\boldsymbol{x}^{\text{masked}}$. Notice that the golden tokens are obtained by feeding the original image to the image tokenizer encoder.

In order to generate corrupted image samples $\mathcal{I}^{\text{corrupted}}$ for the enhancer, we sample tokens' replacements from the BEiT output distribution $p_{\text{BEiT}}$ at each masked position $j$ of the encoded $\boldsymbol{x}^{\text{masked}}$:

$$
\begin{aligned}
x_j^{\text{sampled}} &\sim p_{\text{BEiT}}(x_j^{\text{sampled}} \mid \boldsymbol{x}^{\text{masked}}), \ \ \text{for } j \in \boldsymbol{m}, \\
\boldsymbol{x}^{\text{corrupted}} &= \text{replace}(\boldsymbol{g}, \boldsymbol{m}, \boldsymbol{x}^{\text{sampled}}),
\end{aligned}
\tag{2}
$$

where the $\text{replace}(\boldsymbol{g}, \boldsymbol{m}, \boldsymbol{x}^{\text{sampled}})$ operation denotes using the sampled visual token $\boldsymbol{x}^{\text{sampled}}$ to replace golden tokens of $\boldsymbol{g}$ at positions $\boldsymbol{m}$. Next, the image tokenizer decoder maps $\boldsymbol{x}^{\text{corrupted}}$ to a corrupted image $\mathcal{I}^{\text{corrupted}}$. The whole image tokenizer is frozen (*i.e.*, not updated throughout the pre-training phase), which directly uses the publicly available[3] pre-trained DALL-E dVAE weight (Ramesh et al., 2021) following BEiT.

The enhancer takes the corrupted image $\mathcal{I}^{\text{corrupted}}$ as input. For the RESPIX visual pretext task, the enhancer is optimized by a combination of $l_1$ and $l_2$ loss for pixel regression. For the REVDET variant, the enhancer is learned by binary cross-entropy loss for replaced visual token detection. The gradients of the enhancer are not back-propagated through the generator.

In this paper, we study CIM self-supervised pre-trained vanilla ViT (Dosovitskiy et al., 2020) and vanilla ResNet (He et al., 2016) models. The vanilla ViT models refer to the design from (Dosovitskiy et al., 2020; Touvron et al., 2021a) without further architectural change such as using relative position embeddings (Shaw et al., 2018) and LayerScale (Touvron et al., 2021b). The vanilla ResNet-50 model refers to the torchvision ResNet-50 (Paszke et al., 2019) without any architectural change. The larger ResNet-50x2 and ResNet-50x4 models follows the canonical design in SimCLR (Chen et al., 2020b). We conduct experiments on $16\times$ or $32\times$ V100 GPUs with 32GB memory.

---

[3] https://github.com/openai/DALL-E

## A.4 PRE-TRAINING & FINE-TUNING CONFIGURATIONS

### A.4.1 THE IMAGENET-1K CIM PRE-TRAINING CONFIGURATIONS FOR VANILLA VIT AND RESNET MODELS

| Pre-training Config. (ViT & ResNet) | Value |
| --- | --- |
| Optimizer | AdamW (Loshchilov & Hutter, 2017) |
| Pre-training Epochs | 300 |
| Peak Learning Rate | 1.5e-3 |
| Batch Size | 2048 |
| Weight Decay | 0.05 |
| Optimizer Momentum $(\beta_1, \beta_2)$ | (0.9, 0.98) (Vaswani et al., 2017) |
| Learning Rate Schedule | Cosine Decay |
| Gradient Clipping | 3.0 |
| Warmup Epochs | 10 |
| # Masked Patches for the Generator | 100 to 120, Random Masking |
| The Generator's Depth | 4 to 6 |
| The Generator's Width | Same to the Enhancer (ViT), 384 (ResNet) |
| The Enhancer's Loss Weight | 1 for REVDET, 10 for RESPIX |
| Data Augmentation | RandomResizedCrop Only |
| Dropout (Srivastava et al., 2014) | ✗ |
| Stochastic Depth (Huang et al., 2016) | ✗ |
| LayerScale (Touvron et al., 2021b) | ✗ |
| Pos. Emb. in Transformer Layers | 1-D Absolute Pos. Emb. (Dosovitskiy et al., 2020) |
| Patch Size | 16 |
| Pre-training Resolution | 224 |

Table 13: **The ImageNet-1K CIM pre-training settings for vanilla ViT-S/16, ViT-B/16 and ResNet-50 models.** Notably, the pre-training configurations are almost the same for different architectures. We implement the pre-training using the codebase of BEiT (Bao et al., 2021). Mixed precision and deepspeed acceleration are used.

### A.4.2 THE IMAGENET-1K IMAGE CLASSIFICATION FINE-TUNING CONFIGURATIONS FOR VANILLA VIT MODELS

| Fine-tuning Config. (ViT) | Value |
|---|---|
| Optimizer | AdamW (Loshchilov & Hutter, 2017) |
| Fine-tuning Epochs | 200 for ViT-S/16, 100 for ViT-B/16 |
| Peak Learning Rate | 3e-3 for ViT-B/16 RESPIX, 5e-3 for ViT-B/16 REVDET, 3e-3 or 4e-3 for ViT-S/16 |
| Layer-wise Learning Rate Decay (Bao et al., 2021) | 0.8 (Clark et al., 2020) |
| Batch Size | 1024 |
| Weight Decay | 0.05 |
| Optimizer Momentum $(\beta_1, \beta_2)$ | (0.9, 0.999) |
| Learning Rate Schedule | Cosine Decay |
| Warmup Epochs | 5 |
| Gradient Clipping | ✗ |
| Dropout (Srivastava et al., 2014) | ✗ |
| Stochastic Depth (Huang et al., 2016) | 0.1 |
| Label Smoothing (Szegedy et al., 2016) | 0.1 |
| Mixup (Zhang et al., 2017) | 0.8 |
| CutMix (Yun et al., 2019) | 1.0 |
| Random Augmentation (Cubuk et al., 2020) | 9 / 0.5 |
| Patch Size | 16 |
| Fine-tuning Resolution | 224 |
| Test Resolution | 224 |
| Test Crop Ratio | 0.95 |
| Loss Function | Cross Entropy Loss |

Table 14: **The ImageNet-1K image classification fine-tuning recipes for vanilla ViT-S/16 and ViT-B/16.** We implement the fine-tuning using the codebase of BEiT (Bao et al., 2021). Mixed precision and deepspeed acceleration are used. We select the best learning rate out of {3e-3, 4e-3, 5e-3} for different sized models and pre-training objectives, and the absolute difference between the worst and the best learning rate is less than 0.3 in terms of the top-1 accuracy.

A.4.3    THE IMAGENET-1K IMAGE CLASSIFICATION FINE-TUNING CONFIGURATIONS FOR
VANILLA RESNET-50

| Fine-tuning Config. (ResNet-50) | 100 Epoch FT | 300 Epoch FT | 600 Epoch FT |
|---|---|---|---|
| Optimizer | | AdamW (Loshchilov & Hutter, 2017) | |
| Peak Learning Rate | | 12e-3 | |
| Layer-wise Learning Rate Decay (Bao et al., 2021) | | ✗ | |
| Batch Size | | 2048 | |
| Learning Rate Schedule | | Cosine Decay | |
| Loss Function | | Binary Cross Entropy Loss | |
| Warmup Epochs | | 5 | |
| Weight Decay | 0.02 | 0.02 | 0.01 |
| Fine-tuning Resolution | 160 | 224 | 224 |
| Test Resolution | | 224 | |
| Test Crop Ratio | | 0.95 | |
| Repeated Augmentation (Berman et al., 2019; Hoffer et al., 2019) | ✗ | ✓ | ✓ |
| Random Augmentation (Cubuk et al., 2020) | 6 / 0.5 | 7 / 0.5 | 7 / 0.5 |
| Mixup (Zhang et al., 2017) | 0.1 | 0.1 | 0.2 |
| CutMix (Yun et al., 2019) | | 1.0 | |
| Label Smoothing (Szegedy et al., 2016) | 0.1 | ✗ | 0.1 |
| Stochastic Depth (Huang et al., 2016) | ✗ | ✗ | 0.05 |
| Dropout (Srivastava et al., 2014) | | ✗ | |
| Layer-wise Learning Rate Decay | | ✗ | |

Table 15: **The ImageNet-1K image classification fine-tuning recipes for vanilla ResNet-50.** We use the AdamW optimizer. The hyperparameter settings basically follows (Wightman et al., 2021). We implement the fine-tuning based on the codebase of BEiT (Bao et al., 2021). Mixed precision and deepspeed acceleration are used. For other self-supervised baseline approaches we compared in Table 2, we select the best learning rate out of {5e-3, 8e-3, 12e-3} and keep other settings unchanged.

