# OpenReview forum: "Corrupted Image Modeling for Self-Supervised Visual Pre-Training"
_ICLR.cc/2023/Conference — ICLR 2023 notable top 25%_

### Official Review · Reviewer_wVPU · 2022-10-25

**Confidence:** 4
**Correctness:** 3
**Technical Novelty And Significance:** 3
**Empirical Novelty And Significance:** 3
**Recommendation:** 6

**Clarity, Quality, Novelty And Reproducibility:**

Clarity
- The writing is clear.

Quality
- It requires more explanations and comparisons to emphasize the strengths compared to existing methodologies to improve, as I mentioned above.

Novelty
- The technical contribution is incremental, but the introduced approach is worth sharing.

Reproducibility
- The code is provided, but some components are missing, e.g., model architecture for the proposed method ("CNNForMaskedImageModeling")

**Strength And Weaknesses:**

Strengths
- The writing is clear and easy to understand.
- The proposed method is a generic self-supervised learning scheme in a Masked Image Modeling manner, which can be applied to CNN and ViT.
- Extensive experimental results demonstrate the effectiveness of the proposed method.

Weaknesses
- The proposed method has a weakness in that a pre-trained model is necessarily required in any form (e.g., Dall-E models) as a part of the generator.
- In the case of the ViT model, I am not sure if the proposed method is a better way than the existing BeiT or MAE-style MIM approaches. Also, related explanations and comparisons are lacking. For example, the comparison with MAE (83.1% in Table 1) will be fair when the MAE generator is used instead of external Dall-E models (82.6% in Table 10). In this case, MAE shows better performance than the proposed method.
- The explanation of the need for a generator is somewhat lacking. Is it not possible to replace this by applying strong augmentations (e.g., cutout, ColorJitter, etc.) only to random patches?
- I think the main novelty comes from that made MIM pre-training in CNN. Is there any unique advantage of MIM pre-training in CNN?
- Missing comparison; BeiT and MAE have also known for their superior performances on ADE20K semantic segmentation. However, the authors only report the ADE20K performances for CNNs. Could the proposed method achieve better performances on the ADE20K than BeiT or MAE?

**Summary Of The Paper:**

This manuscript proposes Corrupted Image Modeling, which is a self-supervised learning framework for generic architectures, e.g., CNN and ViT. Specifically, the proposed method reconstructs or predicts the original image from a generated image whose are partially reconstructed from [MASK] tokens. The extensive experiments showed the effectiveness of the proposed method on various vision tasks, such as image classification and semantic segmentation.

**Summary Of The Review:**

Overall, I think the paper is worth sharing, although I have several concerns, as mentioned above. I hope the authors will address the issues and questions raised.

---

> ### Author Response · Authors · 2022-11-18
> **Thanks and Response to Reviewer wVPU**
>
> Thanks for your valuable comments and your appreciation for our technical contributions. Below we discuss the points you have raised in detail.
>
> ---
>
> > **Q1: The proposed method has a weakness in that a pre-trained model is necessarily required in any form (e.g., Dall-E models) as a part of the generator.**
>
> We partially agree with that.
> From another perspective, exploiting the auxiliary generator is also a **feature** of our work, which relaxes the architectural constraint of MIM and enable us to pre-train model other than ViTs such as ConvNets.
> Moreover, using different generators could inject different features / properties into the enhancer, which is also an interesting future research direction.
>
> ---
>
> > **Q2: the comparison with MAE (83.1% in Table 1) will be fair when the MAE generator is used instead of external Dall-E models (82.6% in Table 10). In this case, MAE shows better performance than the proposed method.**
>
> The unsatisfactory result of MAE as the generator is mainly due to stochastic sampling is performed on the RGB domain, only some **low-level** features (mainly color) can be changed and corrupted.
> Therefore the enhancer only learns to correct low-level attributes.
>
> In CIM, the reason to use DALL-E dVAE tokenizer is to naturally perform stochastic sampling on replaced **high-level** semantic tokens, not to leverage the external data & knowledge from DALL-E.
> To show that, we re-train a dVAE image tokenizer on ImageNet-1K data only without using additional data, and then we use this tokenizer to pre-train a ViT-B/16 model with RevDet pretext task.
> The fine-tuning performance on ImageNet-1K and ADE20K remains the same:
>
> | Method | ImageNet-1K Top-1 Acc. | ADE20K mIoU |
> |:----|:----:|:----:|
> | DALL-E tokenizer | 83.3  | 43.6 |
> | Our tokenizer | 83.3  | 43.5 |
>
> This is also true for BEiT[1], please refer to Appendix C in BEiT[1] for details.
>
> Lastly, CIM is an analogy of ELECTRA[2] in language modeling.
> For reference, there is an excellent paper[3] on why and how sampling from a MLM output distribution matters for ELECTRA.
>
>
> *References:*
>
> *[1] Hangbo Bao, Li Dong, Songhao Piao, Furu Wei. "BEiT: BERT Pre-Training of Image Transformers." In ICLR, 2022.*
>
> *[2] Kevin Clark, Minh-Thang Luong, Quoc V. Le, Christopher D. Manning. "ELECTRA: Pre-training Text Encoders as Discriminators Rather Than Generators." In ICLR, 2020.*
>
> *[3] Yaru Hao, Li Dong, Hangbo Bao, Ke Xu, Furu Wei. "Learning to Sample Replacements for ELECTRA Pre-Training." In ACL, 2021.*
>
> ---
>
> > **Q3: The explanation of the need for a generator is somewhat lacking. Is it not possible to replace this by applying strong augmentations (e.g., cutout, ColorJitter, etc.) only to random patches?**
>
> For a detailed discussion on artificial corruption *v.s.* CIM, please refer to our General Response 1 (https://openreview.net/forum?id=09hVcSDkea&noteId=a-vE3wgmN9Z).
>
> ---
>
> > **Q4: Is there any unique advantage of MIM pre-training in CNN?**
>
> In addition to better fine-tuning performance, CIM is the first working solution to demonstrate a general, unified, and non-Siamese framework that is capable of pre-training both strong ViT and CNN visual encoders.
>
> Despite ConvNets preceded ViTs in computer vision research, the progress of pre-training ConvNet as well as other hierarchical architectures operating on 2D regular grids now **lag behind** ViTs.
>
> We hope CIM can bring new perspectives to the study of (generative) self-supervised visual pre-training, and encourage the community for exploring more flexible & unified visual representation learning approaches of various architectures.
>
> ---
>
> > **Q5: Could the proposed method achieve better performances on the ADE20K than BeiT or MAE?**
>
> In Table 3 (a) of our paper, we do make some comparisons on ADE20K among ViTs.
> We use a lightweight linear layer as the segmentation head to better reflect the pre-trained representation.
> Here we also add the result of MAE ViT-Base:
>
> | Method | Pre-training Epochs | mIoU |
> |:----:|:----:|:----:|
> | DINO | 1600  | 43.0 |
> | BEiT | 300  | 43.2 |
> | MAE  | 1600  | 43.5 |
> | CIM-ResPix | 300 | 43.5 |
> | CIM-RevDet  | 300  | 43.6 |
>
> We show for ViTs, CIM is also performant when transferred to the semantic segmentation task.
>
> ---
> We would be more than happy to discuss any further questions.
>
> ---

---

> > ### Comment · Reviewer_wVPU · 2022-11-21
> > **After rebuttal**
> >
> > Most of the author's responses have addressed my concerns quite well.
> > Overall, I think this paper is worth sharing, and my score will remain at a 6 as my original review.

---

### Official Review · Reviewer_JSbL · 2022-10-25

**Confidence:** 4
**Correctness:** 4
**Technical Novelty And Significance:** 2
**Empirical Novelty And Significance:** 2
**Recommendation:** 5

**Clarity, Quality, Novelty And Reproducibility:**

The paper is clearly written and easy to understand. The proposed approach is somewhat novel and reproducible by the details provided in the paper.

**Details Of Ethics Concerns:**

n/a.

**Strength And Weaknesses:**

Strength:
* The task of unifying self-supervised learning approach for ViT and CNN is important
* The approach avoids using [MASK] tokens on pretrained models
* Results on ImageNet classification and semantic segmentation are good using either ViT or CNN
* Ablations show the efficacy of different design choices. The discussion of tokenizer overhead is informative.

Weakness:
* At a higher level, it seems to me that CIM is very similar to MIM. Both require the model to learn to reconstruct the masked/corrupted tokens from the surrounding context.
* The gain over BEiT is small in Table 1 and 3. Since CIM depends on a small BEiT generator, I wonder what the real advantage of the proposed approach over BEiT is, especially when the ViT backbone is used.
* For ResNet, I wonder if there can still be a (BEiT-like) Masked Image Modeling baseline to help us understand the benefits of CIM for CNN. All entries in Table 2 are contrastive based.
* The proposed approach requires a generator (small BEiT) to produce the corrupted image, which makes the system more complicated and reliant on another SSL model.
* The generative and discriminative objective (RESPIX, REVDET) are both similarly effective. Is there any further boost when you combine both? It’d be great to have one best choice at the end.
* In Table 9, it seems the benefits of scaling are not very significant e.g. 1 point from ViT-Base to ViT-Large. It may be useful to investigate this further whether it’s caused by suboptimal fine-tuning recipe, generator/tokenizer choice, etc.
* [Option] flalI am curious if CIM-REVDET would do better than CIM-RESPIX and BEiT on ImageNet linear eval or KNN eval since it uses a discriminative objective.



**Summary Of The Paper:**

This paper proposes a self-supervised visual pretraining approach called corrupted image modeling. The idea is to reconstruct the original image pixels or predict whether each token is corrupted or not. The resulting presentation shows good results on ImageNet classification and ADE20K segmentation using either ViT or CNN.

**Summary Of The Review:**

The paper proposes an interesting self-supervised learning approach to model corrupted images to show competitive performance on classification and segmentation. However, in my opinion, the gains over existing approaches are not significant enough to justify the higher system complexity. In addition, it seems to me the proposed CIM is similar to MIM at a higher level.

---

> ### Author Response · Authors · 2022-11-18
> **Thanks and Response to Reviewer JSbL**
>
> Thanks for your valuable comments and your appreciation for our technical contributions. Below we discuss the points you have raised in detail.
>
> ---
> > **Q1: At a higher level, it seems to me that CIM is very similar to MIM. Both require the model to learn to reconstruct the masked/corrupted tokens from the surrounding context.**
>
> We agree with you.
> Both MIM and CIM belong to the generative pre-training family.
> But to our knowledge, CIM is the first working generative pre-training solution for both ViTs & ConvNets.
>
> ---
> > **Q2: I wonder what the real advantage of the proposed approach over BEiT is, especially when the ViT backbone is used.**
>
> We agree with you that there are no obvious advantages for our approach when using ViT.
> Compared with BEiT, the biggest advantage for CIM is that it has no architectural constraints and is also applicable to ConvNets.
>
> ---
> > **Q3: For ResNet, I wonder if there can still be a (BEiT-like) Masked Image Modeling baseline to help us understand the benefits of CIM for CNN. All entries in Table 2 are contrastive based.**
>
> To our knowledge, CIM is the first performant generative pre-training approach for ConvNets. Moreover, CIM also encourages and inspires some recent follow-up works on generative pre-training for ConvNets.
>
> ---
> > **Q4: The proposed approach requires a generator (small BEiT) to produce the corrupted image, which makes the system more complicated and reliant on another SSL model.**
>
> Measure by pre-training time cost, CIM can achieve better performance on both ViTs and ConvNets compared with strong competitors with smaller complexity.
> Please refer to General Response 2 (https://openreview.net/forum?id=09hVcSDkea&noteId=Rlyk_aNXuen) for details.
>
> ---
>
> > **Q5: The generative and discriminative objectives (RESPIX, REVDET) are both similarly effective. Is there any further boost when you combine both? It’d be great to have one best choice at the end.**
>
> We tried directly combining these two objectives for pre-training, and we found the ImageNet-1K fine-tuning performance suffers from a 0.3% degeneration.
>
> We speculate the main reason is we use a simple linear layer (as the decoder) for the enhancer (as the encoder) to learn two distinct pre-training objectives, *i.e.*, the **discriminative** RevDet and the **generative** ResPix, which aren't cooperative. Thus the top layers of the enhancer have to waste their capacity on modeling two distinct pretext tasks rather than learning transferable representations for downstream tasks.
>
> One possible solution is to pre-train a deeper enhancer and only use its top-12 layers. Therefore the layers for modeling specific pretext tasks are dropped during downstream transfer. We tried pre-train a 14-layer ViT-B enhancer and only transfer the top-12 layers, a performance improvement was observed. This is also true for MAE that the MAE decoder size would affect the encoder's downstream preference as shown in its paper.
>
> Finding the optimal decoder size for CIM is correlated with the type of pretext tasks and has to make costly trial-and-error assessments on downstream tasks. We don't consider this as a core contribution of CIM therefore we just use a simple linear layer.
>
> ---
>
> > **Q6: In Table 9, it seems the benefits of scaling are not very significant e.g. 1 point from ViT-Base to ViT-Large. It may be useful to investigate this further whether it’s caused by suboptimal fine-tuning recipe, generator/tokenizer choice, etc.**
>
> Indeed, we only pre-train ViT-Large once with CIM considering our budgets and large model pre-training complexity.
> Multiple trials or searching for a better recipe could help.
> But we believe the main issue is that currently CIM is not a quite scalable pre-training approach.
> Nevertheless, we hope our efforts can encourage the community to discover a better generative pre-training approach that is architectural agnostic, and it seems our efforts have paid off.
>
> Despite ConvNets preceded ViTs in computer vision research, the progress of pre-training ConvNet as well as other hierarchical architectures operating on 2D regular grids now **lag behind** ViTs.
> We hope CIM can bring new perspectives to the study of (generative) self-supervised visual pre-training, and encourage the community for exploring more flexible & unified visual representation learning approaches of various architectures.
>
> ---
> > **Q7: I am curious if CIM-REVDET would do better than CIM-RESPIX and BEiT on ImageNet linear eval or KNN eval since it uses a discriminative objective.**
>
> We do not observe significant advantages in linear eval when using CIM-RevDet objective, as shown below.
>
> | Method | ImageNet-1K Linear Acc. |
> |:----|:----:|
> | BEiT-Base | 56.7 |
> | ResPix-Base | 55.8 |
> | RevDet-Base | 57.0 |
>
> ---
> We would be more than happy to discuss any further questions.
>
> ---

---

> > ### Comment · Reviewer_JSbL · 2022-11-30
> > **Thanks for the feedback.**
> >
> > Thank you for the feedback. After carefully considering the author response and other reviews, I agree it's a meaningful problem to make generative-style pre-training work for both ResNet and ViT, and ideally other new architectures in the future. I also agree with reviewer E2vC that the intuition of why CIM is better in the case of ResNet is not very clear, the performance deltas are small compared to the added training complexity (not computational resource use), and the scalability can use more study.
> >
> > The comparison with artificial corruption is quite interesting. Thanks for preparing that. There it seems the benefits of CIM over zoom-in may not be statistically significant over multiple runs. I wonder if a mix of different artificial corruption techniques can achieve similar effect as CIM. Moreover, there seems to be no ablation study on simple MIM + ResNet with an up-to-date training recipes and well-tuned masking strategy (let me know if I miss it). It'd be important to see if there's a clear gain of CIM over MIM for ResNet, since that's the main contribution of the paper (per author's response to my questions). Given the evidences we have so far, I would like to keep the original rating.

---

> > > ### Author Response · Authors · 2022-12-06
> > > **CIM bests MIM on ResNet**
> > >
> > > The Appendix E of SimMIM (https://arxiv.org/pdf/2111.09886.pdf) conducts a study on MIM + ResNet50x4 (a very big ResNet with 375M parameters, ~14x bigger than ResNet-50). The fine-tuning top-1 accuracy on ImageNet-1K is 81.6. While CIM can achieve 82.6 top-1 accuracy with a similar pre-training and fine-tuning schedule.
> > >
> > >
> > > | Model: ResNet-50x4 (#params: 375M) | Pre-training Epochs | Fine-tuning Epochs | ImageNet-1k Top-1 Acc. |
> > > |:------:|:-------------------:|:---:|:---:|
> > > | From Scratch | - | 400 | 80.9 |
> > > | SimCLR  | 1000  | 100 | 82.6 |
> > > | SimMIM | 300 | 100 | 81.6 |
> > > | **CIM**  | 100  | 100 | **82.2** |
> > > | **CIM**  | 300  | 100 | **82.6** |
> > >
> > >
> > > These results are also summarized in Table 8 of our paper.
> > >
> > > We also notice that the citation of SimCLR in our manuscript is wrong, and we will fix it in the revision.

---

### Official Review · Reviewer_1bnH · 2022-10-27

**Confidence:** 4
**Clarity, Quality, Novelty And Reproducibility:** I think it's a high-quality paper wit…
**Correctness:** 4
**Technical Novelty And Significance:** 4
**Empirical Novelty And Significance:** 3
**Recommendation:** 8

**Strength And Weaknesses:**

## Strengths
- The motivation of the paper makes a lot of sense. Designing more robust and universal pretraining methods that can benefit various model architectures is an important problem in visual representation learning.
- The paper shows high quality in writing and presentation, which makes the readers pretty easy to follow and comprehend.
- The paper explores new generative and discriminative pretext tasks with a single and simple CIM framework. It's very interesting that the discriminative method RevDet, inspired by the ELECTRA in language modeling, shows comparable performance with ResPix. This may provide insights into future research on designing better pretraining objectives.
- Extensive experiments on downstream tasks and datasets clearly demonstrate the effectiveness of the proposed methods and how each component works.

## Weaknesses
- The visualization on ResPix is intuitive and shows how the model learns from the corrupted images and infers the raw image signals. But there is no visualization of RevDet. How well does the model solve this pretext task? Some visualization or discussion will improve the quality of the paper.
- The pertaining objectives of ResPix and RevDet seem independent to me. What if combining them together? Is there any difficulty to do that or do you think this may raise any problems? I'm curious that whether using the generative and discriminative objectives together will lead to better representations.




**Summary Of The Paper:**

The paper explores the problem of self-supervised visual pre-training, which aims to learn better visual representations with unlabeled data. This paper proposes corrupted image modeling (CIM), a new self-supervised learning framework that takes the corrupted and reconstructed images as input, instead of using images with mask tokens. Given the corrupted images, two types of pretext tasks are proposed, i.e., Pixel Residual learning (ResPix) and Replaced Visual token Detection (RevDet). ResPix regresses the corrupted image patches, while RevDet localizes the corrupted patches via binary classification. Experiments on downstream image classification and semantic segmentation tasks demonstrate the effectiveness of the proposed methods.

**Summary Of The Review:**

This paper proposes both new methods and perspectives for the problem of self-supervised visual pre-training. The experiments well support the claims. I think the community will be interested. I would like to give an accept, but the above concerns should be addressed.

---

> ### Author Response · Authors · 2022-11-18
> **Thanks and Response to Reviewer 1bnH**
>
> Thanks for your valuable comments and your appreciation for our technical contributions. Below we discuss the points you have raised in detail.
>
> > **Q1: There is no visualization of RevDet. How well does the model solve RevDet pretext task?**
>
> It is hard to directly visualize RevDet pretext task, since the model learns a discriminative objective.
> But we can quantitatively show how well the model solve this pretext task:
> The final precision of the enhancer is more than 70% / 75% for ViT-Base / -Large when solving the RevDet pretext task respectively, which is similar to ELECTRA (https://github.com/google-research/electra/issues/3).
>
> ---
>
> > **Q2:  What if combining ResPix and RevDet together?**
>
> We tried directly combining these two objectives for pre-training, and we found the ImageNet-1K fine-tuning performance suffers from a 0.3% degeneration.
>
> We speculate the main reason is we use a simple linear layer (as the decoder) for the enhancer (as the encoder) to learn two distinct pre-training objectives, *i.e.*, the **discriminative** RevDet and the **generative** ResPix, which aren't cooperative. Thus the top layers of the enhancer have to waste their capacity on modeling two distinct pretext tasks rather than learning transferable representations for downstream tasks.
>
> One possible solution is to pre-train a deeper enhancer and only use its top-12 layers. Therefore the layers for modeling specific pretext tasks are dropped during downstream transfer. We tried pre-train a 14-layer ViT-B enhancer and only transfer the top-12 layers, a performance improvement was observed. This is also true for MAE that the MAE decoder size would affect the encoder's downstream preference as shown in its paper.
>
> Finding the optimal decoder size for CIM is correlated with the type of pretext tasks and has to make costly trial-and-error assessments on downstream tasks. We don't consider this as a core contribution of CIM therefore we just use a simple linear layer.
>
> ---
>
> We would be more than happy to discuss any further questions.
>
>
> ---

---

> > ### Comment · Reviewer_1bnH · 2022-11-23
> > **Final comments**
> >
> > Thank you for the detailed response.
> > > The final precision of the enhancer is more than 70% / 75% for ViT-Base / -Large when solving the RevDet pretext task respectively, which is similar to ELECTRA (https://github.com/google-research/electra/issues/3).
> >
> > This final precision and discussion should be added to the paper. It could be a useful metric to understand how well the model learns with such a discriminative objective, just like some recent papers use the MIM reconstruction loss as the indicator.
> >
> > > We tried directly combining these two objectives for pre-training, and we found the ImageNet-1K fine-tuning performance suffers from a 0.3% degeneration......
> >
> > The result and discussion should be added to the paper to give readers a bigger picture.
> >
> > Overall, I am satisfied with the response and I think this paper is worth sharing. Thus I keep my original rating.

---

> > > ### Author Response · Authors · 2022-11-23
> > > **Thanks and response**
> > >
> > > Thanks for your valuable comments. We will add the suggested contents in the revision.

---

### Official Review · Reviewer_hKbV · 2022-10-28

**Confidence:** 3
**Correctness:** 3
**Technical Novelty And Significance:** 3
**Empirical Novelty And Significance:** Not applicable
**Recommendation:** 8

**Clarity, Quality, Novelty And Reproducibility:**

The paper is well-written and well-explained, apart from the concerns expressed in the previous section. The idea builds heavily on top of prior works that are smartly composed together. Overall, the technique can be considered novel.

Many implementation details are provided in the main paper and in the appendix. The code is provided as supplementary material. The libraries are specified in a `requirements.txt` file, with pinned version for critical packages (e.g. torch). There is a README.md with precise commands to pre-train and fine-tune the model. Precise commands/instructions to reproduce all the reported results seems to be missing, although all the code seems to be present. Overall, the paper appears to be reproducible, and the authors encourage reproducibility.


**Strength And Weaknesses:**

Strengths:
- The paper builds on top of BEiT to define pretext tasks with plausible images. Dealing with plausible images instead of masked tokens should intuitively improve the quality of the extracted features.
- The paper introduces a general non-Siamese framework for visual pre-trained that achieves compelling results.
- The experiments are extensive with many ablation studies.

Weaknesses & Questions:
- In Table 1 and Table 2 the reported results are the median of 3 independent runs.  Are the competitors using the same convention? Why the median instead of the mean and std? Do all three independent runs yield comparable performance?  It is unclear if that is the same in all the other tables, or if the other tables report the performance of a single run.
- Table 6 proves that sliding window normalization is beneficial to the performance of the proposed method. I did not understand if the same normalization procedure is applicable also to the competitors. I think the paper would benefit from either: (1) specifying why the sliding normalization is not applicable to the competitors or (2) a comparison with the best-performing competitor to exclude most of the performance gains are caused by the normalization procedure.

I am willing to raise my score, especially if my doubts regarding the median performance are unfounded.

**Summary Of The Paper:**

The paper introduces a self-supervised visual pretraining technique called CIM: Corrupted Image Modeling. The main idea is to randomly select patches and replace them with plausible alternatives, instead of using a MASK token. An enhancer network tries to solve pretext tasks on these augmented images (generating all original pixels or classifying the sample into augmented/non-augmented). Once trained, the enhancer can be used as a visual encoder for downstream tasks. Extensive results and ablation studies validate the proposed technique.
Moreover, the paper introduces a sliding window normalization procedure (Fig. 4) that improves performance.


**Summary Of The Review:**

The paper introduces a novel technique for visual pre-training that achieves compelling results. Overall, the experiments are convincing and the ablation extensive. The method appears to be general and useful for many downstream tasks.

---

> ### Author Response · Authors · 2022-11-18
> **Thanks and Response to Reviewer hKbV**
>
> Thanks for your valuable comments and your appreciation for our technical contributions. Below we discuss the concerns that you have raised in detail.
>
> ---
>
> > **Q1: In Table 1 and Table 2 the reported results are the median of 3 independent runs. Are the competitors using the same convention?**
>
> We follow [1] to report the median of different independent runs.
> People tend to report their best results if they have multiple trials.
> We believe it is OK for ImageNet-1K classification and ADE20K segmentation, as they have relatively small results fluctuations.
> Some famous works of MIM such as BEiT, MAE, and SimMIM all report one number for each entry.
> Nevertheless, we encourage the community to report results' statistics if multiple runs are available.
>
>
> *References:*
>
> *[1] Alexander Kirillov, Yuxin Wu, Kaiming He, Ross Girshick. "PointRend: Image Segmentation as Rendering." In CVPR, 2020.*
>
> ---
>
> > **Q2: Why the median instead of the mean and std?**
>
> We follow [1] to report the median of 3 independent runs.
> We believe there are two main reasons:
> - 3 trials are insufficient to report mean and std, while more trials are infeasible for us (pre-training + fine-tuning is very time- & resource-consuming).
> - You cannot release a checkpoint that can reproduce the mean performance, but you do have a checkpoint that corresponds to the median accuracy.
>
>
> *References:*
>
> *[1] Alexander Kirillov, Yuxin Wu, Kaiming He, Ross Girshick. "PointRend: Image Segmentation as Rendering." In CVPR, 2020.*
>
> ---
>
> > **Q3: Do all three independent runs yield comparable performance?**
>
> We believe so.
> As stated in Section 3 of our manuscript, for ImageNet-1K, we observe ~0.2 (±0.1) Top-1 acc. fluctuations, for ADE20K, we observe ~0.5 (±0.25) mIoU fluctuations.
> To our knowledge, these fluctuations not only appear in our study, but also in other approaches like DeiT, BEiT, MAE (*e.g.*, https://github.com/facebookresearch/mae/issues/30), *etc*.
> Overall, ImageNet-1K classification and ADE20K segmentation are widely used reliable benchmarks as they have relatively small results fluctuations.
>
>
> ---
>
> > **Q4: It is unclear if that is the same in all the other tables, or if the other tables report the performance of a single run.**
>
> We report ablation results in other tables with a single run as in most literatures.
> It is too time-consuming and infeasible to have multiple runs (independent pre-training + fine-tuning trials) for each entry in the ablation study.
>
>
> ---
>
>
> > **Q5: Is the same normalization procedure also applicable to the competitors?**
>
> A large-scale MIM study[1] reports 0.3 top-1 accuracy improvement on SwinV2-Large with SimMIM pre-training using our sliding window normalization (see Section 2.4 of [1]).
> This is a non-trivial improvement since SwinV2-Large / SimMIM are state-of-the-art architecture / pre-training approach, respectively.
> We also observe a 0.2 point improvement on MAE-Base with 400 epoch pre-training using our sliding window normalization in our experiment.
>
>
> *References:*
>
> *[1] Zhenda Xie, Zheng Zhang, Yue Cao, Yutong Lin, Yixuan Wei, Qi Dai, Han Hu. "On Data Scaling in Masked Image Modeling." arXiv preprint arXiv:2206.04664 (2022).*
>
> ---
>
> We would be more than happy to discuss any further questions.
>
>
> ---

---

> > ### Comment · Reviewer_hKbV · 2022-11-22
> > **Thank you for your response**
> >
> > Thank the authors for the satisfactory response. I have no further questions, I am raising my recommendation score since the doubts about the reported metrics in this paper are mostly solved.

---

### Official Review · Reviewer_E2vC · 2022-10-29

**Confidence:** 4
**Correctness:** 3
**Technical Novelty And Significance:** 2
**Empirical Novelty And Significance:** 2
**Recommendation:** 5

**Clarity, Quality, Novelty And Reproducibility:**

The paper needs more rigorous formulations. Zero equations in the main paper, and the formulations provided in the appendix are also sparse. It makes it hard to grip the specific details to reproduce the work.

The proposed method is novel in its detailed formulation to use BEiT to corrupt image regions.



**Strength And Weaknesses:**

**Strength:**

1. The method is reasonably motivated at the high level that more sophisticated operators may produce better results than vanilla masking.
2. The method successfully leveraged BEiT to corrupt the image, which is interesting.
3. Experimental results show improvements in downstream tasks for small and base size models.

**Weaknesses:**

1. The enhancer network's reconstruction ability does not seem very good. It is not very clear why making minor changes (mostly smoothing from the figures) can lead to better pretraining. The paper tries to make some arguments, but they are a bit convolved and do not convey a good intuition.
2. The improvements on small and base size models are incremental, especially compared to BEiT. It does not seem to be statistically significant or rewarding enough to justify the complexity of training the model.
3. The results on larger CNN and ViT are not impressive.



**Summary Of The Paper:**

This paper proposes a visual pretraining method that replaces vanilla masking in mask image modeling (MIM) with image corruption. A small BEiT is used to generate the corrupted images. An enhancer network is trained to predict the original image and used as the final pretrained network. The paper explores two training objectives: one to reconstruct the original image and the other to predict where the corrupted region is. Experiments are done with ResNet and ViT on ImageNet-1K and ADE20K. Image classification and semantic segmentation are downstream tasks to test the performance of the pretrained networks.

**Summary Of The Review:**

The paper is moderately novel and clear. The results have encouraging aspects but are not strong overall.

---

> ### Author Response · Authors · 2022-11-18
> **Thanks and Response to Reviewer E2vC**
>
> Thanks for your valuable comments and your appreciation for our technical contributions. Below we discuss the points you have raised in detail.
>
> ---
>
>
> > **Q1: The enhancer network's reconstruction ability does not seem very good. It is not very clear why making minor changes (mostly smoothing from the figures) can lead to better pre-training.**
>
> First of all, we believe better reconstruction ability in pretext task **≠** better transfer learning performance in downstream tasks, *i.e.*, MLM, MIM & CIM are not downstream tasks that we are really interested in.
> [1] shows that a model that performs well on MIM pretext tasks could fail to transfer well.
> Meanwhile, our goal is to find a pretext task that **transfer** well, but not push our models to learn pretext tasks well.
> By now, the best way to evaluate a pretext task's / pre-training objective's effectiveness is to honestly fine-tune the pre-trained models on downstream tasks.
>
>
> Secondly, CIM is not simply smoothing the input images.
> Its corruption patterns are not random choices, but sampled from a trainable BEiT's output distribution.
> Therefore a model pre-trained via CIM is somewhat learning a BEiT pretext task as well, but with less architectural constraint, *i.e*, CIM can be extended to ConvNets.
>
> Quantitatively, we also show CIM outperforms simple random smoothing & erasing as well as several other artificial corruptions in our General Response 1 (https://openreview.net/forum?id=09hVcSDkea&noteId=a-vE3wgmN9Z).
>
> Lastly, CIM is an analogy of ELECTRA[2] in language modeling.
> For reference, there is an excellent paper[3] on why and how sampling from a MLM output distribution matters for ELECTRA.
>
>
> *References:*
>
> *[1] Zhenda Xie, Zheng Zhang, Yue Cao, Yutong Lin, Yixuan Wei, Qi Dai, Han Hu. "On Data Scaling in Masked Image Modeling." arXiv preprint arXiv:2206.04664 (2022).*
>
> *[2] Kevin Clark, Minh-Thang Luong, Quoc V. Le, Christopher D. Manning. "ELECTRA: Pre-training Text Encoders as Discriminators Rather Than Generators." In ICLR, 2020.*
>
> *[3] Yaru Hao, Li Dong, Hangbo Bao, Ke Xu, Furu Wei. "Learning to Sample Replacements for ELECTRA Pre-Training." In ACL, 2021.*
>
>
> ---
>
> > **Q2: It does not seem to be statistically significant or rewarding enough to justify the complexity of training the model.**
>
> For a detailed comparison of pre-training complexity, please refer to General Response 2 (https://openreview.net/forum?id=09hVcSDkea&noteId=Rlyk_aNXuen).
>
>
> ---
>
> > **Q3: The results on larger ConvNets and ViT are not impressive.**
>
> We agree with you this time.
>
>
> ---
> We would be more than happy to discuss any further questions.
>
> ---

---

> > ### Comment · Reviewer_E2vC · 2022-12-03
> > **Thanks**
> >
> > Thank you for the authors' efforts to provide more results and put together the response.
> > I am good with reasoning in Q1. However, for Q2, the experimental results show insignificant differences in model accuracy. I will keep my rating.

---

> > > ### Author Response · Authors · 2022-12-06
> > > **Thank you for your response**
> > >
> > > Thank you for your response. CIM is not perfect. We respect your decision and rating.

---

### Official Review · Reviewer_UGiU · 2022-10-31

**Confidence:** 3
**Correctness:** 3
**Technical Novelty And Significance:** 3
**Empirical Novelty And Significance:** 2
**Recommendation:** 6

**Clarity, Quality, Novelty And Reproducibility:**

This paper is clearly written and in good quality. The whole pipeline is easy to reproduce based on the code provided by the authors.

**Strength And Weaknesses:**

Strength:
1. The analysis about the difference between CIM and MIM and the advantage of CIM is interesting.
2. The authors provide abundant experiments on several datasets to validation the effectiveness of their method.

Weaknesses:
1. How are the special mask embeddings designed? Are they trainable?
2. The authors refer to “golden token” several times before providing its definition in the REVDET paragraph in Sec.2.1. It would be better if the authors can reorganize for this problem.
3. It seems that the generated corrupted images from the pretrained dVAE decoder are somehow the smoothed version of the original images. Therefore I wonder if it is possible to directly produce corrupted image by artificial smoothing on some random regions without training any new modules?


**Summary Of The Paper:**

This paper focuses on the self-supervised learning. Based on masked image modeling, the authors introduce a new algorithm which uses corrupted as training sources instead of masked ones. They utilize a new trainable module including a pretrained transformer decoder to generate such corrupted images. The models are trained to reconstruct the original images or predict whether a patch is corrupted for self-supervision.

**Summary Of The Review:**

The proposed method is generally a good one, with inspiring information on using non-contrasive self-supervised learning on CNN. However I am concerned with the necessity of using an extra module for the generation of corrupted images.

---

> ### Author Response · Authors · 2022-11-18
> **Thanks and Response to Reviewer UGiU**
>
> Thanks for your valuable comments and your appreciation for our technical contributions. Below we discuss the concerns you have raised in detail.
>
> ---
>
>
> > **Q1: How are the special mask embeddings designed? Are they trainable?**
>
> The special mask embedding (*a.k.a.*,`[MASK]`) is a random initialized trainable embedding widely used in masked language modeling (MLM) such as the famous BERT[1], ReBERTa[2], as well as in masked image modeling (MIM) such as BEiT[3], MAE[4].
>
> Specifically, `[MASK]` is used to replace the language tokens in MLM or image patches in MIM that we want to mask out, and the model is required to reconstruct the missing language tokens or image patches at all `[MASK]` positions based on visible / non-masked language tokens or image patches.
>
>
> *References:*
>
> *[1] Jacob Devlin, Ming-Wei Chang, Kenton Lee, Kristina Toutanova. "BERT: Pre-training of Deep Bidirectional Transformers for Language Understanding." In NAACL, 2019.*
>
> *[2] Yinhan Liu, Myle Ott, Naman Goyal, Jingfei Du, Mandar Joshi, Danqi Chen, Omer Levy, Mike Lewis, Luke Zettlemoyer, Veselin Stoyanov. "RoBERTa: A Robustly Optimized BERT Pretraining Approach." arXiv preprint arXiv:1907.11692 (2019)*
>
> *[3] Hangbo Bao, Li Dong, Songhao Piao, Furu Wei. "BEiT: BERT Pre-Training of Image Transformers." In ICLR, 2022.*
>
> *[4] Kaiming He, Xinlei Chen, Saining Xie, Yanghao Li, Piotr Dollár, Ross Girshick. "Masked Autoencoders Are Scalable Vision Learners." In CVPR, 2022.*
>
> ---
>
> > **Q2: The authors refer to "golden token" several times before providing its definition in the REVDET paragraph in Sec.2.1. It would be better if the authors can reorganize for this problem.**
>
> Thanks for your kind reminder.
> In this work, the "golden token" refers to the output visual tokens of the frozen DALL-E dVAE encoder given the same input image as the small trainable BEiT.
> We have refined the definition of "golden token" in the revision.
> For ease of reviewing, we highlight the added or revised text in red color.
>
> ---
>
> > **Q3: It seems that the generated corrupted images from the pre-trained dVAE decoder are somehow the smoothed version of the original images. Therefore I wonder if it is possible to directly produce corrupted images by artificial smoothing on some random regions without training any new modules?**
>
> For a detailed discussion on artificial corruption *v.s.* CIM, please refer to our General Response 1 (https://openreview.net/forum?id=09hVcSDkea&noteId=a-vE3wgmN9Z).
>
> ---
>
> We would be more than happy to discuss any further questions.
>
> ---

---

> > ### Comment · Reviewer_UGiU · 2022-11-21
> > **Thank you for your response**
> >
> > Thank the authors for the detailed response. I have no further questions.

---

### Author Response · Authors · 2022-11-18
**General Response 2: Pre-training Complexity**

We study the relationship between the pre-training time and downstream performances of different approaches for both ViTs and ConvNets.
We show CIM is a performant and **unified** pre-training approach for both ViTs and ConvNets that outperforms strong **architecture-specific** competitors with **less** pre-training time.
This study is also presented in the Appendix of our paper.

Since CIM is built upon BEiT, we choose BEiT as the masked image modeling baseline approach of ViTs. Here, we first study the ViT-B model's 100-epoch fine-tuning performance on ImageNet-1K val set with different pre-training schedules. The wall-clock time of 1-epoch pre-training of CIM is about 1.8x of BEiT (CIM has an additional tokenizer decoder compared with BEiT) on the same machine.

| Method | Pre-training Epochs | Relative Pre-training Time | ImageNet-1k Top-1 Acc. |
|:----:|:----:|:----:|:----:|
| BEiT | 300  | 1.0x | 82.9 |
| BEiT | 800  | 2.7x | 83.2 |
| CIM  | 300  | 1.8x | 83.3 |
| BEiT | 1600 | 5.3x | 83.3 |
| **CIM**  | 800  | 4.8x | **83.4** |

In the following, we also study the ResNet-50x4 model's 100-epoch fine-tuning performance on ImageNet-1K val set with different pre-training schedules. We choose SimMIM [1] as the masked image modeling baseline approach of ConvNets, for it reports the ResNet-50x4 model's result in Appendix E of its paper. The wall-clock time of 1-epoch pre-training of CIM is about 2.6x of SimMIM (CIM has an additional generator, including a small BEiT and a tokenizer encoder & decoder compared with SimMIM) on the same machine.

| Method | Pre-training Epochs | Relative Pre-training Time | ImageNet-1k Top-1 Acc. |
|:------:|:-------------------:|:---:|:---:|
| SimMIM | 300 | 1.0x | 81.6 |
| **CIM**  | 100  | 0.9x | **82.2** |

These results imply that CIM can obtain better fine-tuning performance with less pre-training time compared with baseline approaches, especially for ConvNets.

We observe that the DALL-E tokenizer of CIM is essentially a large ConvNets and adds non-trivial overhead during pre-training. Meanwhile, the recently proposed ViT-VQGAN [2] reports much higher throughputs, which is a promising replacement for the heavy DALL-E tokenizer to further boost the pre-training efficiency of CIM. We will study the effects of other image tokenizers in the future.

---

*References:*

*[1] Zhenda Xie, Zheng Zhang, Yue Cao, Yutong Lin, Jianmin Bao, Zhuliang Yao, Qi Dai, and Han Hu. "Simmim: a simple framework for masked image modeling." In CVPR, 2022.*

*[2] Jiahui Yu, Xin Li, Jing Yu Koh, Han Zhang, Ruoming Pang, James Qin, Alexander Ku, Yuanzhong Xu, Jason Baldridge, and Yonghui Wu. "Vector-quantized image modeling with improved vqgan." In ICLR, 2022.*

---

### Author Response · Authors · 2022-11-18
**General Response 1: Random Artificial Corruption v.s. CIM**

There is a fantastic report[1] studying the transfer learning performance on ImageNet-1K when ViT-16/B is pre-trained with some random **artificial** image corruptions, *i.e.*:

- (a) zoomed-in: the pre-trained model is required to predict / outpaint the content outside the zoomed-in central image region
- (b) zoomed-out: the model is required to predict the region of interest / foreground region given a zoomed-out image.
- (c) distortion: given a fisheye algorithm distorted image, the model is required to reconstruct the original input.
- (d) blur: given an image blurred via a convolution kernel, the model is required to deblur it.
- (e) de-colorization: the input colored image is converted from RGB to grayscale, and the model is required to colorize and recover it.

We also study two additional **artificial** image corruption, *i.e.*:

- (f) random erasing[2]: randomly blurring image rectangle regions and erasing pixels with random values.
- (g) random smoothing: randomly blurring image rectangle regions via Gaussian blur.

Compared with these **artificial** image corruptions, CIM is a kind of **neural** image corruption.
Below we summarize ImageNet-1K top-1 classification accuracy (%) of different models pre-trained via different corruptions (ViT-B/16 pre-trained with 300 epochs):

| Corruptions | Types | ImageNet-1K Top-1 Acc. |
|:----|:----:|:----:|
| (a) zoomed-in | artificial corruption  | 82.7 |
| (b) zoomed-out | artificial corruption  | 82.2 |
| (c) distortion | artificial corruption  | 82.1 |
| (d) blur | artificial corruption  | 81.8 |
| (e) de-colorization | artificial corruption  | 81.4 |
| (f) random erasing | artificial corruption  | 82.1 |
| (g) random smoothing | artificial corruption  | 81.9 |
| (h) **CIM** | **neural** corruption  | **83.3** |

Results of (a)-(e) are directly from Table 1 in [1].
We show that CIM as a kind of **neural** corruption significantly outperforms a series of **artificial** image corruption.

---

*References:*

*[1] Yunjie Tian, Lingxi Xie, Jiemin Fang, Mengnan Shi, Junran Peng, Xiaopeng Zhang, Jianbin Jiao, Qi Tian, Qixiang Ye. "Beyond Masking: Demystifying Token-Based Pre-Training for Vision Transformers." arXiv preprint arXiv:2203.14313 (2022).*

*[2] Zhun Zhong, Liang Zheng, Guoliang Kang, Shaozi Li, Yi Yang. "Random Erasing Data Augmentation." In AAAI, 2020.*

---

### Decision · Program_Chairs · 2023-01-20

**Decision:**

Accept: notable-top-25%

**Justification For Why Not Higher Score:**

Two reviewers had lingering concerns about the experiments

**Justification For Why Not Lower Score:**

Four reviewers liked the paper, and two of them rated it 8.

**Metareview: Summary, Strengths And Weaknesses:**

Six experts reviewed the paper. Four reviewers were positive about the paper, especially post-rebuttal, and two had lingering concerns post-rebuttal. Particularly, reviewers E2vC and JSbL found the results insignificant, but the other reviewers were positive about the paper as a whole and commented that "The experiments well support the claims". Hence, the decision is to recommend the paper for acceptance. The authors are encouraged to make the necessary changes to the paper to the best of their ability following the reviewers' comments.

**Note From Pc:**

if the above contains the word "oral" or "spotlight" please see: "oral" presentation means -> notable-top-5% and "spotlight" means -> notable-top-25%. As stated in our emails, we are disassociating presentation type from AC recommendations